# *Artemisia annua* L. Polyphenol-Induced Cell Death Is ROS-Independently Enhanced by Inhibition of JNK in HCT116 Colorectal Cancer Cells

**DOI:** 10.3390/ijms22031366

**Published:** 2021-01-29

**Authors:** Eun Joo Jung, Anjugam Paramanantham, Hye Jung Kim, Sung Chul Shin, Gon Sup Kim, Jin-Myung Jung, Chung Ho Ryu, Soon Chan Hong, Ky Hyun Chung, Choong Won Kim, Won Sup Lee

**Affiliations:** 1Department of Biochemistry, Institute of Health Sciences, Gyeongsang National University School of Medicine, Jinju 52727, Korea; eunjoojung@gnu.ac.kr; 2Department of Internal Medicine, Institute of Health Sciences, Gyeongsang National University Hospital, Gyeongsang National University School of Medicine, Jinju 52727, Korea; anju.udhay@gmail.com; 3Research Institute of Life Science, College of Veterinary Medicine, Gyeongsang National University, Jinju 52828, Korea; gonskim@gnu.ac.kr; 4Department of Pharmacology, Institute of Health Sciences, Gyeongsang National University School of Medicine, Jinju 52727, Korea; hyejungkim@gnu.ac.kr; 5Department of Chemistry, Research Institute of Life Science, Gyeongsang National University, Jinju 52828, Korea; sshin@gnu.ac.kr; 6Department of Neurosurgery, Institute of Health Sciences, Gyeongsang National University Hospital, Gyeongsang National University School of Medicine, Jinju 52727, Korea; gnuhjjm@gnu.ac.kr; 7Department of Food Technology, Research Institute of Life Science, Gyeongsang National University, Jinju 52828, Korea; ryu@gnu.ac.kr; 8Department of Surgery, Institute of Health Sciences, Gyeongsang National University Hospital, Gyeongsang National University School of Medicine, Jinju 52727, Korea; hongsc@gnu.ac.kr; 9Department of Urology, Institute of Health Sciences, Gyeongsang National University Hospital, Gyeongsang National University School of Medicine, Jinju 52727, Korea; kychung@gnu.ac.kr

**Keywords:** *Artemisia annua* L., polyphenols, cell death, ROS, JNK, colorectal cancer, SP600125, p53

## Abstract

c-Jun N-terminal kinase (JNK) is activated by chemotherapeutic reagents including natural plant polyphenols, and cell fate is determined by activated phospho-JNK as survival or death depending on stimuli and cell types. The purpose of this study was to elucidate the role of JNK on the anticancer effects of the Korean plant *Artemisia annua* L. (pKAL) polyphenols in p53 wild-type HCT116 human colorectal cancer cells. Cell morphology, protein expression levels, apoptosis/necrosis, reactive oxygen species (ROS), acidic vesicles, and granularity/DNA content were analyzed by phase-contrast microscopy; Western blot; and flow cytometry of annexin V/propidium iodide (PI)-, dichlorofluorescein (DCF)-, acridine orange (AO)-, and side scatter pulse height (SSC-H)/DNA content (PI)-stained cells. The results showed that pKAL induced morphological changes and necrosis or late apoptosis, which were associated with loss of plasma membrane/Golgi integrity, increased acidic vesicles and intracellular granularity, and decreased DNA content through downregulation of protein kinase B (Akt)/β-catenin/cyclophilin A/Golgi matrix protein 130 (GM130) and upregulation of phosphorylation of H2AX at Ser-139 (γ-H2AX)/p53/p21/Bak cleavage/phospho-JNK/p62/microtubule-associated protein 1 light chain 3B (LC3B)-I. Moreover, JNK inhibition by SP600125 enhanced ROS-independently pKAL-induced cell death through downregulation of p62 and upregulation of p53/p21/Bak cleavage despite a reduced state of DNA damage marker γ-H2AX. These findings indicate that phospho-JNK activated by pKAL inhibits p53-dependent cell death signaling and enhances DNA damage signaling, but cell fate is determined by phospho-JNK as survival rather than death in p53 wild-type HCT116 cells.

## 1. Introduction

Colorectal cancer is one of the major causes of cancer-related deaths worldwide. Reactive oxygen species (ROS) are generally higher in colorectal cancer cells than normal control cells, and studies of multiple mechanisms by ROS are still needed for better strategies to treat colorectal cancer [1,2]. Natural polyphenols, present in many fruits and vegetables, exert not only ROS-scavenging abilities but also ROS-scavenging independent actions and are involved in mitochondrial processes and mitochondrially triggered cell death [3]. Since the anticancer activity of polyphenols is more effective and less toxic to cancer patients than conventional radiotherapy and chemotherapy, many natural polyphenols are attracting new therapeutic targets [4]. Polyphenols can be classified into four groups: phenolic acids (e.g., caffeic acid and cinnamic acid derivative), flavonoids (e.g., kaempferol, quercetin, and luteolin), polyphenolic amides (e.g., capsaicin), and other polyphenols (e.g., resveratrol and curcumin) [2]. Natural flavonoids have been shown to attenuate the incidence and recurrence of colorectal cancer through their antiperoxidative, antioxidant, and anti-inflammatory effects [5].

Mitogen-activated protein kinases (MAP kinases) are involved in the regulation of a wide range of biological processes in response to many kinds of stimuli including growth factors and stresses [6]. Four major subfamilies of the MAP kinase family contain extracellular regulated kinase 1/2 (ERK1/2), c-Jun N-terminal kinase (JNK), p38, and ERK5 [6]. Studies using MAP kinase inhibitors have shown that they play a role in the regulation of apoptosis, gene expression, mitosis, differentiation, and immune response [6]. The JNK subfamily consists of JNK1, JNK2, and JNK3, and their multiple isoforms are generated through alternative splicing [7]. SP600125 is well known as a selective and reversible inhibitor of JNK kinases [6]. Many internal and external stimuli cause excess levels of ROS generation, leading to oxidative DNA damage and activation of JNK signaling [8]. Phosphorylation of H2AX at Ser-139 (γ-H2AX), a marker for DNA damage, can be induced by JNK as well as Ataxia-telangiectasia mutated (ATM), Ataxia-telangiectasia and Rad3-related (ATR), and DNA-dependent protein kinase (DNA-PK) associated with DNA damage-regulated signaling [9,10]. JNK activation is known to play an important role in cell death caused by the *Annurca* apple polyphenol extract in MDA-MB-231 breast cancer cells [11]. p62/SQSTM1 (p62), also known as sequestosome-1, plays a role in the regulation of proteasome inhibitor-induced autophagy in human retinal pigment epithelial cells [12]. LC3B-II is a lipidated form of microtubule-associated protein 1 light chain 3B (LC3B)-I, and the conversion of LC3-I to LC3-II is usually detected by an induction of autophagy [13,14]. Autophagic cell death can occur via the inhibition of JNK-mediated p62 expression and the increase in autophagy markers such as Beclin-1 and LC3B-II depending on an autophagy-inducing condition and cell types [14,15,16,17]. Moreover, JNK is known to induce apoptosis, necroptosis, ferroptosis, and pyroptosis and to play a role in cell survival by positively regulating JNK-mediated autophagy depending on cell stimuli [18,19].

*Artemisia annua* L., commonly known as sweet wormwood, has been used as tea and press juice to treat malaria, fever, and chills in Asia and Africa, and artemisinin is a sesquiterpene lactone compound isolated from the Chinese plant *Artemisia annua* [20,21]. Artemisinin and its bioactive derivatives showed potent anticancer effects in various human cancer cells by controlling ROS, DNA damage, DNA repair, cell cycle arrest, cell death, inflammatory response, angiogenesis, and multiple signal transduction pathways [20,21,22]. It is known that JNK activation is not involved in the inhibition of tumor necrosis factor (TNF)-α-induced inflammatory response by artemisinin in Hep3B hepatocarcinoma cells [22], whereas the inhibition of lipopolysaccharide-induced JNK activation by artemisinin enhances p40 subunit production of interleukin-12 in RAW264.7 macrophage cells [23]. Studies on the anticancer activities of *Artemisia annua* L. were mainly accomplished by artemisinin and its derivatives. However, little is known about the role of JNK on the anticancer effects of *Artemisia annua* L. polyphenols, especially in colorectal cancer cells.

Recently, we reported that the tumor suppressor p53 enhanced the anticancer effects of polyphenols isolated from the Korean plant *Artemisia annua* L. (pKAL) in HCT116 colorectal cancer cells [24]. In this study, we further investigated the anticancer effects and mechanisms caused by pKAL in p53 wild-type HCT116 cells. Thus, we found that pKAL induced morphological changes and necrosis or late apoptosis through the regulation of multiple cell survival and death signaling pathways including JNK activation. Moreover, our results revealed that phospho-JNK activated by pKAL played a role in cell survival by inhibiting p53-dependent cell death signaling in p53 wild-type HCT116 cells.

## 2. Results

### 2.1. Effect of pKAL on the Regulation of Cell Morphology in p53 Wild-Type HCT116 Colorectal Cancer Cells

To further elucidate the anticancer mechanisms by pKAL, we observed morphological changes according to the increase in pKAL concentration for 48 and 72 h in p53 wild-type HCT116 cells using a phase-contrast microscopy. As shown in Figure 1A, cell–cell contact was significantly reduced by pKAL treatment for 48 h in a concentration-dependent manner (compare the areas indicated with arrows in panels a’–d’). Compared to the dimethyl sulfoxide (DMSO)-treated control, HCT116 cell morphology altered by pKAL was different depending on the amount treated with pKAL: at 25 µg/mL pKAL treatment, most cells were slightly enlarged while maintaining cell–cell interactions, and some cells were changed into small round or slightly larger irregular shapes; the 50 µg/mL pKAL treatment resulted in more decrease in cell number and cell–cell interaction than 25 µg/mL pKAL, and small round or irregularly shaped cells were more increased than 25 µg/mL pKAL; at 100 µg/mL pKAL treatment, most cells turned into small round-shaped cells (cells 1–3, panel d”) or deformed-shaped cells containing large aggregated vesicles indicated by arrows (cells 4 and 5, panel d”) (Figure 1A). Moreover, cell morphology was more significantly changed by pKAL treatment for 72 h; especially, 100 µg/mL of pKAL treatment resulted in the increase in certain secreted vesicles or degraded cell remnants in the extracellular region, indicated by arrows (Figure 1B, panel d’). These results suggest that pKAL-induced anticancer mechanisms may differ depending on the amount treated with pKAL.

### 2.2. Effect of pKAL on the Regulation of PI Uptake, Apoptosis, ROS, and Acidic Vesicles

To investigate whether pKAL-induced death is associated with apoptosis, necrosis, ROS production, and acidic vesicles, we performed flow cytometric analysis of annexin V/propidium iodide (PI)-, dichlorofluorescein (DCF)-, and acridine orange (AO)-stained HCT116 cells treated with pKAL for 48 h. PI is a red fluorescent dye that can bind to nuclear DNA through the permeabilized plasma membrane of dying or dead cells but not live cells [25]. The integrity of plasma and nuclear membrane is decreased during necrotic and late apoptotic cell death processing, and thus, PI can pass the permeabilized plasma membrane and can bind nuclear DNA of necrotic and late apoptotic cells but not early apoptotic cells [26,27]. In this study, the annexin V/PI staining results showed that the cell population stained by PI was significantly increased by pKAL treatment in a concentration-dependent manner, 0.1% DMSO (12.56%), 25 µg/mL pKAL (16.00%), 50 µg/mL pKAL (28.20%), and 100 µg/mL pKAL (64.13%), suggesting an increase in necrotic cells associated with loss of plasma membrane integrity (Figure 2A, upper left section). Moreover, the cell population co-stained by annexin V and PI was also significantly increased by pKAL in a concentration-dependent manner, 0.1% DMSO (0.32%), 25 µg/mL pKAL (9.36%), 50 µg/mL pKAL (17.67%), and 100 µg/mL pKAL (27.64%), suggesting an increase in late apoptotic cells associated with loss of plasma membrane integrity (Figure 2A, upper right section). However, the DCF-staining results showed that ROS-producing cells were increased significantly at 25 µg/mL pKAL (33.53%) compared to the 0.1% DMSO control (5.91%) but slightly decreased at 50 µg/mL pKAL (27.32%) and greatly decreased at 100 µg/mL pKAL (5.09%) (Figure 2B). It is known that the intracellular acidic organelles and autophagy can be detected by AO red fluorescence staining [28]. In this study, the AO staining results showed that acidic vesicle-producing cells were notably increased by pKAL treatment in a concentration-dependent manner: 0.1% DMSO (5.01%), 25 µg/mL pKAL (12.57%), 50 µg/mL pKAL (33.20%), and 100 µg/mL pKAL (56.15%) (Figure 2C). These findings suggest that pKAL induces necrotic or late apoptotic cells associated with loss of plasma membrane integrity and increase in acidic vesicles regardless of ROS generation.

### 2.3. Effect of pKAL on the Regulation of DNA Content, Cell Cycle, and Intracellular Granularity

Nuclear DNA fragmentation by various nucleases is one of the hallmarks of apoptosis. Thus, apoptotic cells contain a broad hypodiploid DNA that can be detected as a sub-G1 peak in DNA histogram analysis [29,30]. In addition, early apoptotic cells have an increased granularity and/or cell density due to probably condensed chromatin and cytoplasm, while late apoptotic and necrotic cells have a decreased intracellular granularity due to probably a leakage of cell contents through ruptured plasma membranes [31]. To better understand pKAL-induced cell death mechanisms, we performed DNA content analysis using flow cytometry in HCT116 cells treated with pKAL of 25 and 50 µg/mL for 72 h. As expected, the cell population in the G0/G1 and G2/M phases was significantly decreased by pKAL treatment in a concentration-dependent manner compared to the DMSO-treated control, resulting in the increased cell population in sub-G1 peak (M1 gate): 0.1% DMSO (3.83%), 25 µg/mL pKAL (26.16%), and 50 µg/mL pKAL (51.86%) (Figure 3, DNA histograms of bottom panels). Moreover, the intensity of side scatter (SSC-H) in the G0/G1, S, and G2/M but not sub-G1 phases was significantly increased by pKAL treatment in a concentration-dependent manner comparted to the DMSO-treated control (Figure 3, dot plots of top panels). These results suggest that pKAL induces apoptotic cell death associated with a decrease in DNA content and an increase in intracellular granularity.

### 2.4. pKAL-Induced Cell Death Mechanisms in p53 Wild-Type HCT116 Cells

To further elucidate pKAL-induced cell death mechanisms, we performed Western blot analysis on signaling proteins related to cell survival and death in HCT116 cells treated with different amounts of pKAL for 72 h. The results showed that protein levels of Akt, β-catenin, cyclophillin A, and GM130 were significantly downregulated by the long-term treatment of pKAL in a concentration-dependent manner (Figure 4A), whereas the protein levels of γ-H2AX, p53, 21, Bax, cleaved Bak/caspase-3, phospho-JNK, p62, and LC3B-I but not of Beclin-1 and LC3B-II were upregulated by pKAL (Figure 4B,C). However, except for the high-molecular-weight-modified forms of GM130, γ-H2AX, LC3B-I, and LC3B-II, most of the proteins regulated by pKAL appeared to be degraded by unknown proteases activated under the condition of massive cell death by 100 µg/mL pKAL. These findings indicate that the anticancer effects of pKAL are associated with downregulation of Akt/β-catenin/cyclophillin A survival signaling, loss of GM130-related Golgi integrity, γ-H2AX-related DNA damage induction, activation of p53/p21/Bax/Bak/caspase-3 apoptotic signaling, and upregulation of phospho-JNK/p62/LC3B-I autophagy-related signaling.

### 2.5. Effect of the ROS Inhibitor NAC on Morphological Changes and Protein Levels Altered by pKAL

To investigate whether ROS is involved in morphological changes and cell death caused by pKAL, we performed phase-contrast microscopy and Western blot analysis after 25 µg/mL pKAL treatment for 36 h in HCT116 cells. As shown in Figure 5A, cell morphology was significantly changed to small round cells with or without cell fragmentation by pKAL treatment but pKAL-induced morphological changes were not significantly prevented by the ROS scavenger *N*-acetyl-L-cysteine (NAC) (Figure 5A). Moreover, the Western blot analysis results showed that the upregulation of phospho-JNK and p62 by pKAL was significantly downregulated by co-treatment of NAC, indicating that JNK-dependent signaling is activated by pKAL in an ROS-dependent manner (Figure 5B). However, the upregulation of p53 and p21, but not LC3B-I and LC3B-II, by pKAL was further increased by the co-treatment of NAC in an unknown mechanism (Figure 5B). These results suggest that pKAL-induced cell death may not be significantly inhibited by NAC in p53 wild-type HCT116 colorectal cancer cells because p53-dependent cell death signaling activated by pKAL is enhanced by the co-treatment of NAC.

### 2.6. Effect of the JNK Inhibitor SP600125 on the Regulation of ROS, DNA Conformational Change, and Acidic Vesicles Induced by pKAL

To elucidate the role of ROS-dependent phospho-JNK associated with pKAL-induced cell death, HCT116 cells were grown for 36 h with 50 µg/mL pKAL treatment in the absence or presence of 2 µg/mL SP600125 and analyzed by flow cytometry after DCF and AO staining. The DCF staining results showed that the ROS-producing cell population was not significantly affected by SP600125 treatment but that it was notably increased by pKAL treatment: compare the hyper-DCF-stained cells between the DMSO control (2.05%), SP600125 alone (5.11%), and pKAL alone (30.83%). However, the increase in the ROS-producing cell population by pKAL was not significantly influenced by the JNK inhibitor SP600125: compare the hyper-DCF-stained cells between pKAL alone (30.83%) and pKAL/SP600125 together (31.25%) (Figure 6A). In Figure 5B, we showed that JNK phosphorylation was occurred by pKAL treatment in an ROS-dependent manner. Therefore, our results suggest that ROS produced by pKAL plays a role upstream of JNK signaling and that ROS-activated phospho-JNK is not involved in an additional ROS production. However, the AO staining results showed that the cell population of AO green- and red-stained cells was significantly increased by pKAL treatment, and this phenomenon was further enhanced by co-treatment of SP600125: compare the AO green-stained cells between pKAL alone (33.73%) and pKAL/SP600125 together (51.87%) and compare the AO red-stained cells between pKAL alone (30.52%) and pKAL/SP600125 together (46.89%). These results indicate that ROS-activated phospho-JNK inhibits pKAL-induced DNA conformational change and acidic vesicle production in an ROS-independent manner, suggesting a survival role of pKAL-induced phospho-JNK in p53 wild-type HCT116 cells.

### 2.7. Effect of SP600125 on the Regulation of Morphological CHANGES, Apoptosis, and PI UpTake Caused by pKAL

To investigate the role of ROS-activated phospho-JNK on the morphological changes and cell death caused by pKAL, HCT116 cells were treated with 50 µg/mL pKAL for 60 h in the absence or presence of 2 µg/mL SP600125 and then analyzed by phase-contrast microscopy. As shown in Figure 7A, the cell morphology was significantly changed to a small round shape (cells 1 and 2, panel c’) or enlarged shape (cells 3–6, panel c’) by pKAL treatment. In addition, the morphological changes induced by pKAL were altered by co-treatment of SP600125, resulting in a decrease in small round cells, an increase in more enlarged cells (cells 1 and 2, panel d’), and an induction of abnormal large vesicles (cells indicated by arrows, panel d’) (Figure 7A). To reveal whether pKAL-induced cell death is inhibited or enhanced by JNK inhibition, we performed flow cytometry after annexin V/PI-staining. The results demonstrated that pKAL-induced apoptosis and PI uptake were enhanced by the co-treatment of SP600125: compare the annexin V-stained cells between pKAL alone (16.36%) and pKAL/SP600125 together (22.17%) and compare the PI-stained cells between pKAL alone (26.96%) and pKAL/SP600125 together (34.38%) (Figure 7B). These results suggest that JNK inhibition by SP600125 but not the induction of small round cells enhances pKAL-induced cell death associated with cellular structural changes and loss of plasma membrane integrity.

### 2.8. Mechanisms Related to the Enhancement of pKAL-Induced Cell Death by SP600125

To examine the mechanisms associated with the enhancement of pKAL-induced cell death by JNK inhibition, HCT116 cells were treated with 50 µg/mL pKAL for 60 h in the absence or presence of 2 µg/mL SP600125 followed by Western blot analysis. Consistent with the results of Figure 4, phospho-JNK, p62, LC3B-I, γ-H2AX, p53, p21, and Bak cleavage were upregulated by pKAL treatment; however, the upregulation of phospho-JNK, p62, LC3B-I, and γ-H2AX by pKAL was decreased by the co-treatment of SP600125 (Figure 8A) whereas the upregulation of p53, p21, and Bak cleavage by pKAL was increased by the co-treatment of SP600125 (Figure 8B). These results indicate that the increase in pKAL-induced cell death by JNK inhibition is related to the activation of p53-dependent cell death signaling. Moreover, our results suggest that pKAL-induced phospho-JNK is involved in both cell survival and death signaling by inhibiting p53-dependent cell death signaling and by activating DNA damage signaling, but cell fate is determined to cell survival rather than death by pKAL-induced phospho-JNK.

### 2.9. Effect of Twice Sequential Treatments of SP600125 on the Regulation of DNA Conformational Change, Acidic Vesicles, Apoptosis, and PI Uptake Caused by pKAL

To further prove the enhancement of pKAL-induced cell death by JNK inhibition, HCT116 cells were long-term treated with 50 µg/mL pKAL for 84 h in the absence or presence of twice sequential treatments of 2 µg/mL SP600125 and then analyzed by flow cytometry of AO- and annexin V/PI-stained cells. The AO staining results showed that the AO green- and red-stained cells were increased by twice sequential treatments of SP600125 compared to the DMSO-treated control and increased by pKAL treatment (Figure 9A). Moreover, the increase in pKAL-induced AO green- and red-stained cells was further enhanced by twice sequential treatments of SP600125: compare the AO green-stained cells between pKAL alone (28.57%) and pKAL/SP + SP together (58.11%) and compare the AO red-stained cells between pKAL alone (11.62%) and pKAL/SP + SP together (50.01%) (Figure 9A). Similarly, the annexin V/PI staining results showed that annexin V- and PI-stained cells were increased by twice sequential treatments of SP600125 compared to the DMSO-treated control and increased by pKAL treatment (Figure 9B). Moreover, the increase in pKAL-induced annexin V- and PI-stained cells was significantly enhanced by twice sequential treatments of SP600125: compare the annexin V-stained cells between pKAL alone (18.15%) and pKAL/SP + SP together (31.76%) and compare the PI-stained cells between pKAL alone (32.11%) and pKAL/SP + SP together (43.34%) (Figure 9B). These results further support that JNK inhibition by SP600125 enhances pKAL-induced cell death by increasing DNA conformational change, acidic vesicles, apoptosis, and loss of plasma membrane integrity.

## 3. Discussion

Although technology for surgery, chemotherapy, radiotherapy, and immunotherapy to treat colorectal cancer continues to improve, colorectal cancer is still the leading cause of death in men and women worldwide [32]. Natural polyphenols derived from plants are known to prevent many chronic diseases, including cancer, cardiovascular disease, and neurodegenerative disorders [33]. Moreover, natural polyphenols were shown to increase sensitization to chemotherapy and radiotherapy in colorectal cells by modulating cytokine and chemokine production, suggesting that polyphenols may play an important role in the treatment of colorectal cancer [34]. To better understand the anticancer mechanisms of *Artemisia annua* L. polyphenols (pKAL) associated with the tumor suppressor p53, we further investigated the anticancer effects and mechanisms of pKAL in p53 wild-type HCT116 human colorectal cancer cells. Thus, our results revealed that JNK inhibition by SP600125 promoted pKAL-induced cell death by activating p53-dependent cell death signaling regardless of ROS in HCT116 cells.

Cell fate can be determined by cell morphology, ROS, metabolism, and intracellular pH, especially in stem cell biology, and ROS upregulation can alter cell morphology and motility through Ras activation [35,36]. Natural compounds are known to induce morphological changes associated with the induction of noncanonical cell death such as paraptosis, mitotic catastrophe, necroptosis, anoikis, and ferroptosis [37]. Common features of apoptotic cells are cell shrinkage, plasma membrane blebbing, cell detachment, nuclear condensation, DNA fragmentation, externalization of phosphatidylserine (PS), and activation of caspases, whereas common features of necrotic cells are organelle swelling and plasma membrane rupture [38]. In this study, our results showed that the morphological changes by pKAL were different depending on the amount of pKAL. Especially, almost all cells were changed to small round cells by 100 µg/mL pKAL treatment without the induction of enlarged cells, which were significantly induced at 50 µg/mL pKAL treatment (Figure 1). In addition, pKAL-induced cell death was higher at 100 µg/mL treatment than 50 µg/mL in an ROS-independent manner (Figure 2A,B), and β-actin was downregulated by 100 µg/mL pKAL treatment but not by 50 µg/mL (Figure 4A, bottom panel). These results suggest that enlarged cells may be more resistant to cell death than small round cells and that the morphological changes by 100 µg/mL pKAL treatment may be due to cytoskeletal changes caused by the downregulation of β-actin during pKAL-induced necrosis or late apoptosis.

Curcumin, a polyphenol isolated from the rhizome of Curcuma longa, is known to exploit anticancer effects through downregulation of the PI3K-Akt-mTOR pathway in head and neck cancer cells [39], and downregulation of the Wnt/β-catenin pathway by Akt inhibition is associated with antimony (Sb)-induced neurotoxicity [40]. In this study, both Akt and β-catenin were significantly downregulated by the long-term treatment of pKAL in a concentration-dependent manner (Figure 4A, first and second panels). Cyclophilin A, a soluble cytosolic peptidyl-prolyl *cis-trans* isomerse, is known as a biomarker of necrotic cell death due to early secretion outside the cell under necroptosis [41]. However, cyclophilin A is also known to inhibit hypoxia/reoxygenation-induced apoptosis in H9c2 cardiomyocytes by inhibiting the expression of NADPH oxidase 2 (Nox2) via the Akt pathway [42], suggesting a differential role of cyclophilin A according to cell status. In this study, cyclophilin A was downregulated by 100 µg/mL of pKAL treatment in an unknown mechanism but not was significantly reduced by 25 and 50 µg/mL of the pKAL treatment (Figure 4A, third panel).

In Figure 2A, our results showed that pKAL-induced cell death was primarily associated with loss of plasma membrane integrity, suggesting that the Golgi integrity may also be affected by pKAL treatment. Thus, we investigated the effect of pKAL on the regulation of GM130, which is known as a *cis*-Golgi matrix protein that plays an importance role in the maintenance of Golgi apparatus [43,44]. Very interestingly, our results showed that GM130 was gradually downregulated by pKAL treatment in a concentration-dependent manner; however, the high-molecular-weight-modified form of GM130 was induced by pKAL in an unknown mechanism and appeared to be very stable under conditions of pKAL-induced necrosis or late apoptosis (Figure 4A, fourth panel).

The anticancer effects of plant polyphenols are associated with the activation of DNA damage signaling [45]. γ-H2AX is caused by many kinds of DNA damage-inducing reagents, and it is also known to occur as a consequence of apoptosis regardless of ROS [9,46]. Many kinds of natural polyphenols, such as curcumin, resveratrol, genistein, luteolin, and quercetin, can stabilize the p53 protein through p53 phosphorylation, p53 acetylation, and reduced oxidative stress in various cancer cell lines [47]. Polyphenol-induced p53 plays a role in the regulation of cell cycle, DNA repair, metabolism, senescence, autophagy, and apoptosis through the activation of its direct and indirect downstream targets in various cancer cell types [47,48]. In this study, our results showed that γ-H2AX was significantly upregulated by pKAL in a dose-dependent manner (Figure 4B, first panel). In addition, p53, p53-dependent targets such as p21 and Bax, and cleavage of Bak and caspase-3 were also significantly upregulated by pKAL treatment (Figure 4B).

Resveratrol, a natural polyphenol found in grapes, protects chronic myelogenous leukemia (CML) cells from stress-induced apoptosis through its antioxidant effects, whereas it promotes autophagic cell death in CML cells via JNK-mediated p62 upregulation [49]. p62 is a stress-inducible protein with multiple domains: A Phox1 and Bem1p (PB1) domain, a ZZ-type zinc finger domain, a tumor necrosis factor (TNF) receptor-associated factor 6 (TRAF6) binding domain, a microtubule-associated protein 1A/1B-light chain 3 (LC3)-interacting region (LIR), a Keap1-interacting region (KIR), and an ubiquitin-associated (UBA) domain [50]. p62 interacts with phagophores through the LIR domain and with the ubiquitinated protein aggregates through the UBA domain and sequesters the target cargo into inclusion bodies via its PB1 domain. Moreover, p62 functions as a signaling hub and an autophagy adaptor [51], and the upregulation and downregulation of p62 have been implicated in tumor formation, cancer promotion, and chemotherapeutic resistance [50]. Tanshinone I, one of the major compounds of *Salvia miltirrhiza*, exerts cytotoxicity associated with p62 accumulation and conversion of the LC3-I soluble form to the LC3-II autophagic vesicle-associated form in malignant pleural mesothelioma cells, and this phenomenon was significantly inhibited by the JNK inhibitor SP600125, indicating that JNK activation is critical for tanshinone I-induced p62-dependent autophagy [52]. In this study, our results showed that p62 was significantly upregulated by pKAL in an ROS-dependent manner similar to phospho-JNK (Figure 5B), and this phenomenon was suppressed by the JNK inhibitor SP600125, suggesting a survival role of p62 as a downstream target of JNK (Figure 8A). Moreover, pKAL-induced cell death was further enhanced by twice sequential treatments of SP600126 by increasing DNA conformational change, acidic vesicles, apoptosis, and loss of plasma membrane integrity (Figure 9). However, the enhancement of pKAL-induced cell death by SP600125 was not due to the induction of small round cells but was significantly associated with cellular structural changes such as the induction of abnormal large vesicles and loss of plasma membrane integrity (Figure 7A).

In conclusion, this study suggests that phospho-JNK/p62 signaling activated by pKAL plays a survival role in p53 wild-type HCT116 cells by inhibiting p53-dependent cell death signaling irrespective of ROS. Research into the sophisticated anticancer mechanisms of natural polyphenols will be a big help in establishing powerful anticancer therapies that can more effectively treat colon cancer. Polyphenols such as flavonoids extracted from *Artemisia annua* L. are known to exhibit antimalarial, anti-proliferative, and anticancer effects through their antioxidant properties [53,54]. In contrast, our results suggest that polyphenols extracted from *Artemisia annua* L. may have anticancer activities in an ROS-independent mechanism by altering the plasma membrane integrity. Therefore, further studies are still needed to elucidate the more efficient cell death mechanism of pKAL in HCT116 colorectal cancer cells.

## 4. Materials and Methods

### 4.1. Reagents

Fetal bovine serum (FBS), penicillin/streptomycin, and trypLE express with phenol red were from Life Technologies (Carlsbad, CA, USA). RPMI 1640 medium was from HyClone (Logan, Utah, USA). SP600125 was from LC Laboratories (Woburn, MA, USA). *N*-acetyl-L-cysteine (NAC), RNase A, 2’,7’-dichlorofluorescein diacetate (DCF-DA), acridine orange (AO), and propidium iodide (PI) were from Sigma (St. Louis, MO, USA). Acrylamide/bis-acrylamide 37.5:1 solution (40%), Tween-20, and DMSO were from Amresco (Solon, OH, USA). Annexin V-Fluos was from Roche (Mannheim, Germany). The dishes, plates, tubes, and pipettes for cell culture were from SPL Life Sciences (Pocheon, Republic of Korea) or Thermo Scientific (Rockford, IL, USA). The Amersham Protran 0.2 µM nitrocellulose (NC) membrane was from GE Healthcare Life Sciences. The ECL Ottimo Western blot detection kit was from TransLab (Daejeon, Republic of Korea). The D-Plus^TM^ ECL Pico System was from DonginLS (Seoul, Republic of Korea). The JNK1/2/3 (phospho-T183/Y185) antibody was from Bioworld (St. Louis Park, MN, USA). The Akt1/2/3 (H-136), cyclophillin A (CyPA, 6-YD13), Bak (G-23), caspase-3 (E-8), JNK1 (D-6), p53 (DO-1), p21 (C-19), Beclin-1 (BECN-1, H-300), and GAPDH (FL-335) antibodies were from Santa Cruz (Santa Cruz, CA, USA). The GM130, Bax, β-catenin, and p62 lck ligand (Sequestosome-1) antibodies were from BD Biosciences (San Jose, CA, USA). The Phospho-H2AX (Ser-139) antibody was from Upstate Biotechnology (Lake Placid, NY, USA). The LC3B (ab51520) antibody was from abcam (Cambridge, MA, USA). The β-actin (clone AC-15) antibody was from Sigma (St. Louis, MO, USA). The secondary goat anti-rabbit and anti-mouse horseradish-peroxidase (HRP) conjugates were from Bio-Rad (Hercules, CA, USA) and Enzo Life Sciences (Plymouth Meeting, PA, USA).

### 4.2. pKAL Components

*Artemisia annua* L., called gaddongsook in Korea, is an annual herb that blooms bright yellow flowers in September and grows to about 2 m in height. The whole tissues of *Artemisia annua* L. were obtained from a gaddongsook farm in Jinju, Korea, and were used in this study. The pKAL compounds were isolated from mixed tissues including lyophilized roots, stems, leaves, and flowers of *Artemisia annua* L. and were identified by liquid chromatography-tandem mass spectrometry (LC/MS/MS) as previously described [24,55]. The pKAL compounds are caffeic acid, quercetin-3-O-galactoside, mearnsetin-glucoside, kaempferol-3-O-glucoside, quercetin-3-O-glucoside, mearnsetin-glucoside, ferulic acid, isorhamnetin-gluoside, diosmetin-7-O-d-glucoside, luteolin-7-O-glucoside, quercetin, quercetagetin-3-O-methyl ether, luteolin, 8-methoxy-kaempferol, quercetagetin-5,3-di-O-methyl ether, kaempferol, 3,5-dihydroxy-6,7,4′-trimethoxyflavone, 3,5-dihydroxy-6,7,3′,4′-tetramethoxyflavone, and isorhamnetin. For the experiment, pKAL compounds were dissolved in DMSO solvent at a concentration of 100 mg/mL and stored in a −20 °C freezer until use.

### 4.3. Cell Culture

The p53 wild-type HCT116 human colorectal cancer cells (KCLB No. 20247) were purchased from Korean cell line bank. The HCT116 cells were maintained in RPMI medium with L-glutamine (300 mg/L), 25 mM HEPES, 25 mM NaHCO_3_, 1% penicillin/streptomycin, and 10% heat-inactivated FBS in a 37 °C incubator supplemented with 5% CO_2_ in a humidified atmosphere. For drug treatment, the cells were split every 3 days and grown in maintenance medium with DMSO control or pKAL treatment for the indicated times.

### 4.4. Phase-Contrast Light Microscopy

Cell morphology was analyzed by phase-contrast light microscopy in a 10× objective (Inf Plan Achro 10× LWD PH, 0.25NA/6.9WD) (EVOS XL Core, Life Technologies) with 150× amplification.

### 4.5. Flow Cytometric Analysis of Annexin V/PI-Stained Cells

Floating and attached cells were collected and incubated with annexin V-Fluos in 10 mM HEPES (pH 7.4)/140 mM NaCl/5 mM CaCl_2_/1 µg/mL PI/phosphate-buffered saline (PBS) solution for 20 min at room temperature (RT), and FL1-H green and FL2-H red fluorescent cells were analyzed by flow cytometric analysis (FACS Calibur, Becton Dickinson).

### 4.6. Flow Cytometric Analysis of DCF-Stained Cells

Floating and attached cells were collected and incubated with 5 µM DCF-DA for 20 min in a 37 °C water bath, and FL-1H green fluorescent cells were analyzed by flow cytometric analysis (FACS Calibur, Becton Dickinson).

### 4.7. Flow Cytometric Analysis of AO-Stained Cells

Acridine orange (AO) staining can be used to detect nuclear chromatin by green fluorescence [56] and to detect acidic organelles and autophagy by red fluorescence [28]. Floating and attached cells were collected and incubated with 1 µg/mL AO/PBS for 20 min in a 37 °C water bath. AO-stained FL1-H green and FL3-H red fluorescent cells were analyzed by flow cytometric analysis (FACS Calibur, Becton Dickinson).

### 4.8. DNA Content Analysis Using Flow Cytometry

Floating cells grown on a 10 cm dish were collected and the attached cells were trypsinized. Both cells were put together, washed with PBS, suspended in 0.5 mL of 0.1% glucose/PBS, and fixed with 5 mL of 70% cold ethanol for at least 2 h at 4 °C. After washing with PBS, the cells were treated with 5 µg/mL PI and 200 µg/mL RNase A in PBS for 30–45 min in a 37 °C water bath. For DNA content analysis, FL2-A red (PI) fluorescent cells were analyzed by flow cytometric analysis (FACS Calibur, Becton Dickinson).

### 4.9. Western Blot Analysis

Whole cells (attached and floating cells) were extracted with an SDS sample buffer and were boiled for 5 min at 95 °C. The resultant proteins were separated using SDS-PAGE and transferred to an NC membrane. The membrane was blocked for 30 min at RT in blocking buffer (3% skim milk, 0.1% Tween-20, and PBS) and then incubated with primary antibody at 4 °C overnight. The blot was then washed with PBST (0.1% Tween-20, PBS) three times for 10 min and incubated with an HRP-conjugated secondary antibody in blocking buffer for 1–2 h. After being washed with PBST, the blot was analyzed with the ECL Western blot detection system.

## Figures and Tables

**Figure 1 ijms-22-01366-f001:**
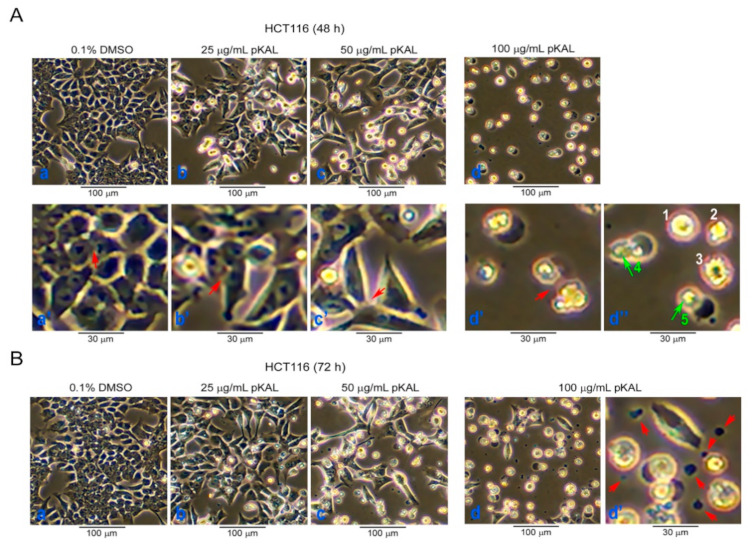
Effect of the Korean plant *Artemisia annua* L. (pKAL) on the regulation of cell morphology in p53 wild-type HCT116 colorectal cancer cells: HCT116 cells were grown with the indicated amount of DMSO or pKAL for 48 h (**A**) and 72 h (**B**) and analyzed by phase-contrast light microscopy. Panels a’–d’ were enlarged from panels a–d, respectively, and panel d’’ was enlarged from panel d.

**Figure 2 ijms-22-01366-f002:**
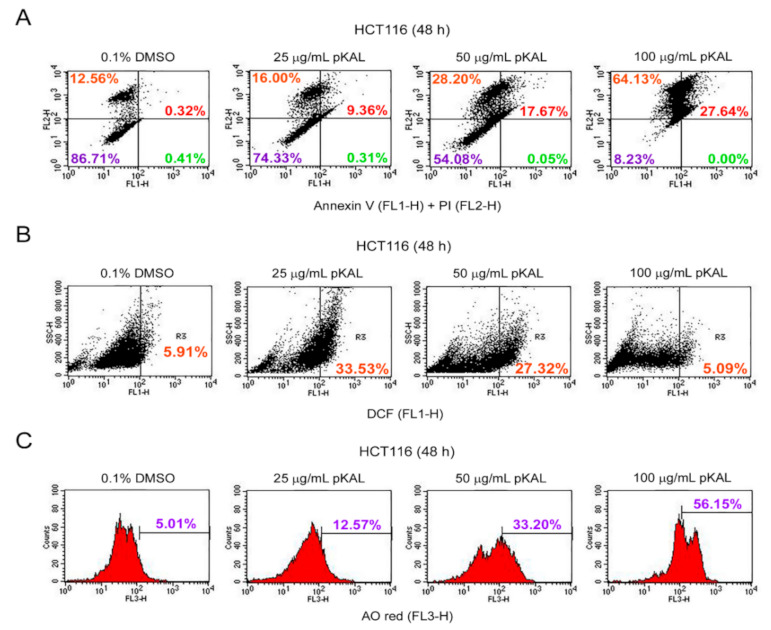
Effect of pKAL on the regulation of propidium iodide (PI) uptake, apoptosis, reactive oxygen species (ROS), and acidic vesicles: HCT116 cells were grown for 48 h with the indicated amount of pKAL. The floating and attached cells were analyzed by flow cytometry: (**A**) co-staining of annexin V (FL1-H) and PI (FL2-H), (**B**) dichlorofluorescein (DCF) (FL1-H) staining, and (**C**) acridine orange (AO) red (FL3-H) staining.

**Figure 3 ijms-22-01366-f003:**
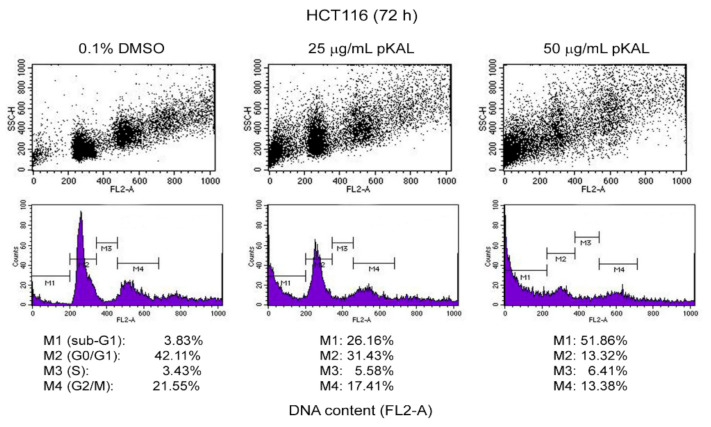
Effect of pKAL on the regulation of DNA content, cell cycle, and intracellular granularity: HCT116 cells were grown for 72 h with the indicated amount of pKAL. The floating and attached cells were analyzed by DNA content analysis using flow cytometry: top panels, dot plots for side scatter (SSC-H) and DNA content (FL2-A); bottom panels, DNA histograms for DNA content (FL2-A).

**Figure 4 ijms-22-01366-f004:**
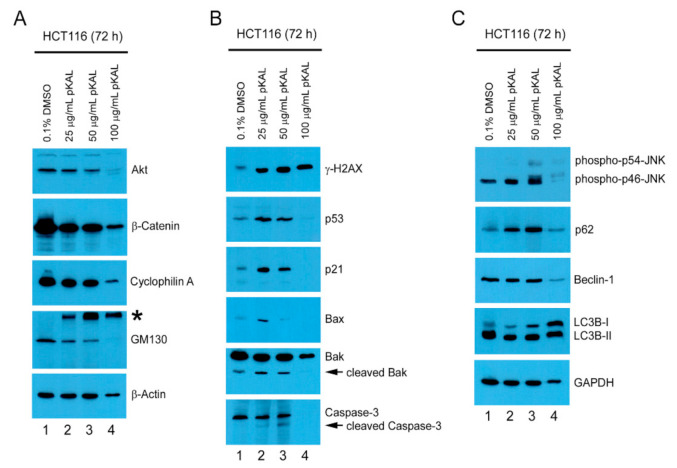
pKAL-induced cell death mechanisms in p53 wild-type HCT116 cells: HCT116 cells were grown on a 10 cm culture dish with DMSO or pKAL treatment for 72 h at the indicated amount. Cells were analyzed by Western blot using the indicated antibodies: (**A**) downregulation of Akt-mediated survival proteins by pKAL. The asterisk indicates the high-molecular-weight post translational modified form of GM130; (**B**) upregulation of apoptosis-related proteins by pKAL. The arrow indicates the low-molecular-weight cleaved form of the corresponding protein; (**C**) upregulation of autophagy-related proteins by pKAL.

**Figure 5 ijms-22-01366-f005:**
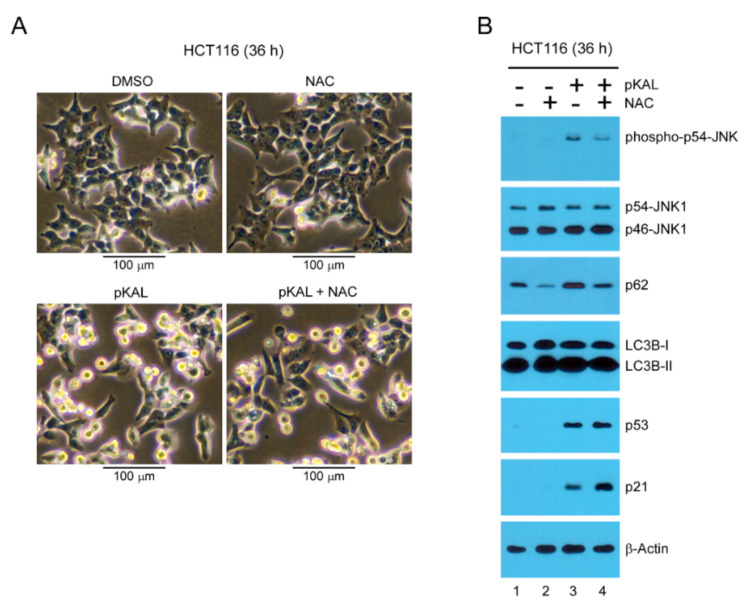
Effect of the ROS inhibitor *N*-acetyl-L-cysteine (NAC) on morphological changes and protein levels altered by pKAL: HCT116 cells were grown for 36 h with 0.1% DMSO, 3 mM of NAC, 25 µg/mL of pKAL, or both 3 mM NAC and 25 µg/mL pKAL in (**A**) phase-contrast light microscopy and (**B**) Western blot analysis.

**Figure 6 ijms-22-01366-f006:**
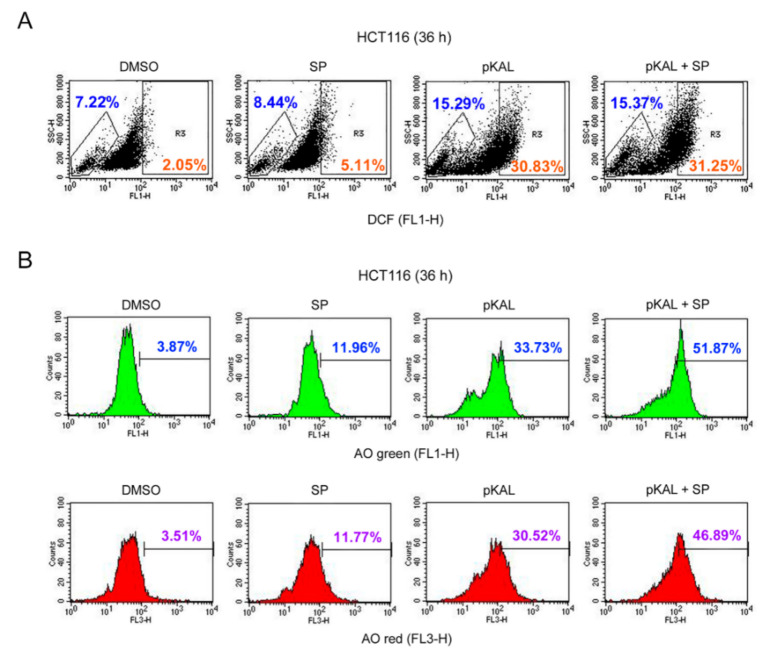
Effect of the c-Jun N-terminal kinase (JNK) inhibitor SP600125 on the regulation of ROS, DNA conformational change, and acidic vesicles induced by pKAL: HCT116 cells were grown with 50 µg/mL pKAL for 12 h, then treated with 2 µg/mL SP600126 (SP), and cultured for an additional 24 h: (**A**) flow cytometric analysis of DCF-stained cells; (**B**) flow cytometric analysis of AO-stained cells.

**Figure 7 ijms-22-01366-f007:**
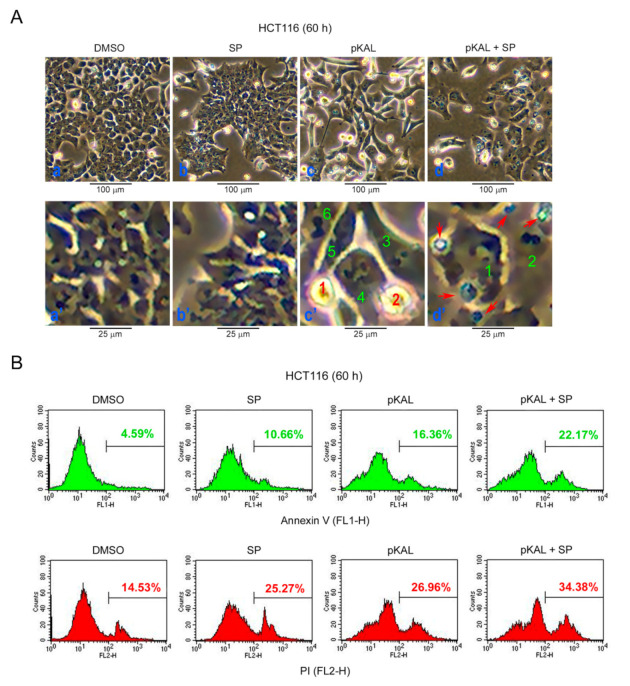
Effect of SP600125 on the regulation of morphological changes, apoptosis, and PI uptake caused by pKAL: HCT116 cells were grown with 50 µg/mL pKAL for 12 h, then treated with 2 µg/mL SP600125, and cultured for an additional 48 h: (**A**) phase-contrast microscopy. Panels a’–d’ were enlarged from panels a–d, respectively, and the numbers in panels c’ and d’ represent morphologically altered cells compared to the control cells in panel a’; (**B**) flow cytometric analysis of annexin V/PI-stained cells.

**Figure 8 ijms-22-01366-f008:**
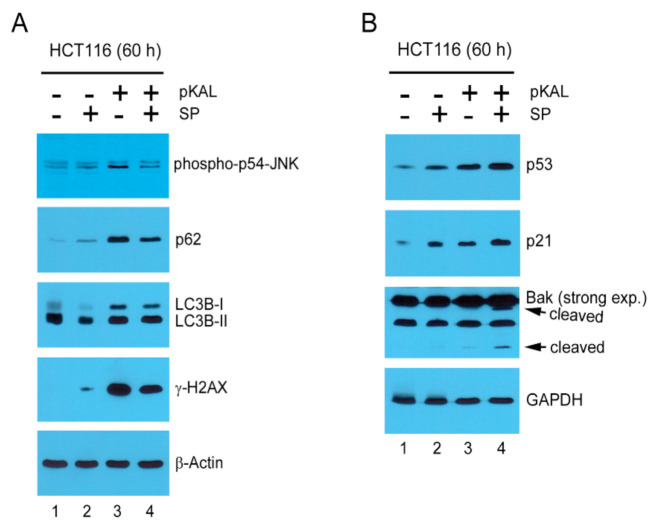
Mechanisms related to the enhancement of pKAL-induced cell death by SP600125: HCT116 cells were grown with 50 µg/mL pKAL for 12 h, then treated with 2 µg/mL SP600125, and cultured for an additional 48 h. Cells were analyzed by Western blot using the indicated antibodies; the arrows represent the low-molecular-weight cleaved forms of Bak: (**A**) downregulated pKAL-dependent targets by the co-treatment of SP600125; (**B**) up regulated pKAL-dependent targets by the co-treatment of SP600125.

**Figure 9 ijms-22-01366-f009:**
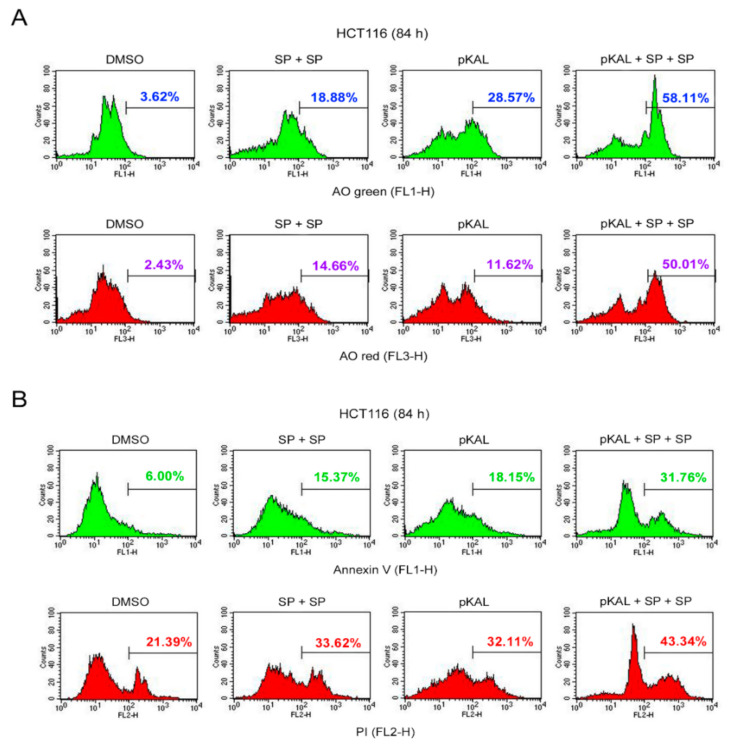
Effect of twice sequential treatments of SP600125 on the regulation of DNA conformational change, acidic vesicles, apoptosis, and PI uptake caused by pKAL: HCT116 cells were grown with 50 µg/mL pKAL for 12 h, treated with 2 µg/mL SP600125 for 24 h, and then once more treated with 2 µg/mL SP for 48 h shown in (**A**) flow cytometric analysis of AO green- and red-stained cells and (**B**) flow cytometric analysis of annexin V/PI-stained cells.

## Data Availability

Not applicable.

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
