# Peer review of "Artemisia annua L. Polyphenol-Induced Cell Death Is ROS-Independently Enhanced by Inhibition of JNK in HCT116 Colorectal Cancer Cells"

_ijms, 2021, doi:10.3390/ijms22031366_

Round 1
Reviewer 1 Report
Some comments are listed below:
The manuscript should be fully revised by a native english speaker. In my opinion, sometimes it is difficult to understand the meaning of the sentences.
In results, the title of each section should not be a summary of the main findings. For example, for section 2.1. I would suggest: “Effect of pKAL on HCT116 colorectal cancer cells morphology”. And a similar comment applies for subsequent sections.
Could the authors explain why they incubated cells with 25 ug/ml of pKAL in the assay described in section 2.5 while the concentration was 50 ug/ml in subsequent sections?
If possible, the discussion section could be completed with results reported in literature for the specific polyphenols found in Artemisia annua L.
The list of references should be fully revised according to the journal style. Quite a lot of them are incomplete (i.e. pages or identifier numbers are missing).
Author Response
Comments and Suggestions for Authors
Some comments are listed below:
1) The manuscript should be fully revised by a native english speaker. In my opinion, sometimes it is difficult to understand the meaning of the sentences.
- Response to comment 1 -
We have partially edited this manuscript during a major revision process. However, if you think that this manuscript should undergo extensive English editing, we will request that this manuscript be fully revised through the English editing service provided by MDPI.
2) In results, the title of each section should not be a summary of the main findings. For example, for section 2.1. I would suggest: “Effect of pKAL on HCT116 colorectal cancer cells morphology”. And a similar comment applies for subsequent sections.
- Response to comment 2 -
The title of each result section has been changed to:
2.1. Effect of pKAL on the regulation of cell morphology in p53 wild-type HCT116 colorectal cancer cells
2.2. Effect of pKAL on the regulation of PI uptake, apoptosis, ROS, and acidic vesicles
2.3. Effect of pKAL on the regulation of DNA content, cell cycle and intracellular granularity
2.4. pKAL-induced cell death mechanisms in p53 wild-type HCT116 cells
2.5. Effect of the ROS inhibitor NAC on morphological changes and protein levels altered by pKAL
2.6 Effect of the JNK inhibitor SP600125 on the regulation of ROS, DNA conformational change, and acidic vesicles induced by pKAL
2.7. Effect of SP600125 on the regulation of morphological changes, apoptosis, and PI uptake caused by pKAL
2.8. Mechanisms related to the enhancement of pKAL-induced cell death by SP600125
2.9. Effect of SP600125 with twice sequential treatments on the regulation of DNA conformational change, acidic vesicles, apoptosis, and PI uptake caused by pKAL
3) Could the authors explain why they incubated cells with 25 ug/ml of pKAL in the assay described in section 2.5 while the concentration was 50 ug/ml in subsequent sections?
- Response to comment 3 -
In section 2.5, we used 25 mg/mL pKAL to investigate the effect of ROS reduction by NAC. Because ROS-producing cells were higher at 25 mg/mL pKAL than 50 mg/mL (Figure 2B). However, pKAL-induced morphological changes, apoptosis, and cell death mechanisms were more significantly altered by 50 mg/mL than 25 mg/mL (Figure1-4). That is why we treated 50 mg/mL pKAL in subsequent studies for the role of JNK inhibition in the regulation of pKAL-induced cell death.
4) If possible, the discussion section could be completed with results reported in literature for the specific polyphenols found in Artemisia annua L.
- Response to comment 4 -
As you suggested, the last part of discussion has been changed to:
In conclusion, this study suggests that phospho-JNK/p62 signaling activated by pKAL plays a survival role in p53 wild-type HCT116 cells by inhibiting p53-dependent cell death signaling irrespective of ROS. Research into the sophisticated anticancer mechanisms of natural polyphenols will be a big help in establishing powerful anticancer therapeutic mechanisms that can more effectively treat colon cancer. Polyphenols such as flavonoids extracted from Artemisia annua L. are known to exhibit antimalarial, anti-proliferative and anticancer effects though their antioxidant properties [53,54]. In contrast, our results suggest that polyphenols extracted from Artemisia annua L. may have anticancer activity in ROS-independent mechanisms by altering plasma membrane integrity. Therefore, further studies are still needed to elucidate the more efficient cell death mechanisms of pKAL in HCT116 colorectal cancer cells.
- Inserted referenes -
[53] Ferreira, J. F. S.; Luthria, D. L.; Sasaki, T.; Heyerick, A. Flavonoids from Artemisia annua L. as Antioxidants and Their Potential Synergism with Artemisinin against Malaria and Cancer. Molecules 2010, 15, 3135-3170.
[54] Skowyra, M.; Gallego, M. G.; Segovia, F.; Almajano, M. P. Antioxidant Properties of Artemisia annua Extracts in Model Food Emulsions. Antioxidants (Basel) 2014, 3, 116-128.
5) The list of references should be fully revised according to the journal style. Quite a lot of them are incomplete (i.e. pages or identifier numbers are missing).
- Response to comment 5 -
As you can see in references section, we carefully revised references according to the IJMS journal style.
Thank you very much for your helpful comments.
Reviewer 2 Report
Thank you very much to the authors for their well-organized research work. They showed well about the topic and did well-planned research. Here I am mentioning some points to consider.
- I think the authors forgot to mention all abbreviations at their first time mentioning. For example, ROS (line 33), PI, DCF, AO, SSC-H/DNA (line 35), Akt, GM130 (line 38, 39), etc.…………. ATM, ATR, DNA-PK (line 70, 71), etc. Some are missing even in the ‘abbreviations’ part. Please make them all similar (ie, if you mention the first time, mention for them all).
- Line 106, What is the meaning of ‘cell-cell contact was significantly reduced’? From the figure, it seems like cells died and dead cells are floating on the media (Figure 1A).
- Line 113, 114, Please enlarge and indicate ‘no clear compartment between the nucleus and cytoplasm with or without large aggregated vesicles’ in the figures.
- Line 117, Please enlarge and indicate ‘certain secreted vesicles or degraded cell remnants’ in the figures.
- Line 200, ‘HCT116 cells’. Line 208, HCT116 colorectal cancer cells! I found this several times. Is there any difference? This study demonstrates that pKAL can cause cancer cell death. What about normal cell? Did you check its effect on normal cells?
- Akt, p-pJNK were checked after 36, 72 hours (Figure 4, 5). What about early time points as these are known to be activated early (ie within 1 or 2 hours or less)?
- Line 212, 3 mM NAC was used as ROS scavenger. Why 3 mM? Was that an effective dose to scavenge ROS in your experimental condition? Please include the data.
- Line 219, 2 mg/mL SP600125. Why 2 mg/mL? If you checked it as an effective dose, please include those data.
- Line 251, increased abnormal aggregated vesicles….. Please enlarge the images and indicate. Do this for all figures wherever you have mentioned special characteristics.
- Line 252, Thus, we couldn't clearly ‘expect’ whether pKAL….. What is the meaning of ‘expect’here? Please revise.
- Line 258, 259, SP600125 enhances pKAL-induced cell death associated with cellular structural changes and loss of plasma membrane integrity, but not by induction of small round What is the meaning of ‘but not by induction of small round cells’? These small round cells seem to be floating dead cells. Did you confirm (Figure 7A)? Does that mean cell death could be induced by forming small round cells? Or the round cells are the fate of the cells after pKAL-induction? One is the reason, another is the result. I think both are different.
- Line 364, in various cancer cell types including A549 non-small ‘cell’ lung cancer cells. Please revise.
- Line 389, SP600125 was not by induction of small round cells, This small round cells seem to be floating dead cells. Does that mean cell death could be induced by forming small round cells? Or it’s the final fate of the pKAL-induced cell?
- Line 419, from where Artemisia annua L. plant roots, stems, leaf, flower were collected?
Round 2
Reviewer 1 Report
The authors have carefully revised the manuscript. Just one minor comment:
Line 414: through?
Author Response
Response to reviewer 1
Line 414: through?
Okay. Thank you!